# Stereotactic Radiosurgery and Stereotactic Fractionated Radiotherapy in the Management of Brain Metastases

**DOI:** 10.3390/cancers16061093

**Published:** 2024-03-08

**Authors:** Sofian Benkhaled, Luis Schiappacasse, Ali Awde, Remy Kinj

**Affiliations:** Department of Radiation Oncology, UNIL-CHUV, Rue du Bugnon 46, 1011 Lausanne, Switzerland

**Keywords:** brain intracranial metastases, stereotactic radiosurgery, stereotactic fractionated radiotherapy, review

## Abstract

**Simple Summary:**

The management of brain metastases (BM) is a major issue in cancer treatment, and one of the main goals of BM treatment is to achieve effective disease control while concurrently preserving neurocognition and quality of life. Stereotactic radiosurgery (SRS) and radiation therapy (SRT) represent a mainstay option that is undergoing a significant paradigm evolution with unexpected opportunities and challenges. This review highlights the evidence and the emerging role of SRS-SRT in patients diagnosed with intact intracranial metastases.

**Abstract:**

The management of brain metastases (BM) remains an important and complex issue in the treatment of cancer-related neurological complications. BM are particularly common in patients diagnosed with lung, melanoma, or breast cancer. Over the past decade, therapeutic approaches for the majority of BM patients have changed. Considering and addressing the fact that patients with BM are living longer, the need to provide effective local control while preserving quality of life and neurocognition is fundamental. Over the past decade, SRS and SRT have become a more commonly chosen treatment option for BM. Despite significant advances in the treatment of BM, numerous questions remain regarding patient selection and optimal treatment sequencing. Clinical trials are critical to advancing our understanding of BM, especially as more therapeutic alternatives become available. Therefore, it is imperative for interdisciplinary teams to improve their understanding of the latest advances in SRS-SRT. This review aims to comprehensively explore SRS and SRT as treatments for BM, covering clinical considerations in their application (e.g., patient selection and eligibility), managing limited and multiple intact BM, addressing brainstem metastases, exploring combination therapies with systemic treatments, and considering the health economic perspective.

## 1. Introduction 

Brain metastases (BM) are ten times more common than primary brain tumors. Without an appropriate treatment, BM progression can lead to neurological morbidity and neurological death [1,2,3]. Lung, breast malignancies and melanoma are the most common causes of BM, accounting for up to 65% of all BM patients [4,5]. The prevalence of BM is increasing for several reasons, including (higher resolution) brain imaging as a cornerstone of cancer staging and surveillance, more effective systemic therapies, and prolonged survival of cancer patients [3,5,6,7]. During the last decade, the therapeutic paradigms for the majority of BM patients have changed significantly [8]. Patients with solitary large, hemorrhagic, symptomatic metastases or those requiring a tissue for histologic diagnosis are currently still candidates for surgery [3,7,9,10]. Despite the fact that neurosurgical resection is an important component of BM management [11], radiation therapy (RT) is the most widely used treatment [3]. Whole brain radiotherapy therapy (WBRT) was introduced in the 1950s and became the most common treatment for patients with multiple BM due to its availability and “simplicity” [2,12]. The rationale for WBRT was based on the hypothesis that since most cancers seed the brain after hematogenous spread, the entire brain could be invaded by micro-metastases [13]. Throughout this period, the prognosis for BM patients was dismal; therefore, little attention was paid to the toxicities of WBRT [12]. In fact, initially, classical chemotherapeutic regimens had almost no efficacy in BM, so almost all patients received local brain-directed therapy [3]. Nowadays, targeted therapies and immune checkpoint inhibitors have strengthened extra- and intracranial control in several malignancies, allowing for more personalized treatment [3,7]. These improvements in the prognosis of BM patients led to evidence of cognitive impairment associated with WBRT, raising concerns regarding its use [14,15]. 

In 1951, Swedish neurosurgeon Lars Leksell pioneered stereotactic radiosurgery (SRS) using an orthovoltage X-ray system and a stereotactic frame to focus the beams on the brain target [16]. The technology was improved, resulting in the clinical appearance of the first Stockholm’s Gamma Knife unit in 1968. Advances in our understanding of BM have led to a number of patient- and disease-specific treatment strategies, including stereotactic radiosurgery (SRS), stereotactic radiotherapy (SRT), hippocampal-avoidance whole-brain radiotherapy (HA-WBRT), surgical resection, and systemic therapies such as targeted therapies and immune checkpoint inhibitors. In contrast to WBRT, SRS combines multiple, highly conformal, convergent beams of high energy to a specific target while sparing adjacent normal tissue [17]. Between the 1990s and 2000s’, SRS-SRT using linear accelerator systems (e.g., Novalis, Brain Lab, CyberKnife, and Accuray) was introduced clinically to treat lesions that were previously untreatable with SRS [17]. Despite the need for rigorous daily quality control, these devices allow for greater treatment flexibility while ensuring precision and accuracy. Today, the data on how to treat BM with SRS-SRT are constantly evolving. SRS-SRT has allowed BM therapy, often in a single fraction, while sparing the adjacent brain. SRS can be delivered in two different forms: either as a single dose (18–24 Gy), or in three fractions of 24–27 Gy or 30–35 Gy in five fractions [3]. An increasing number of local tumors are being treated with Stereotactic Body Radiotherapy (SBRT) instead of surgery, but conclusive evidence of the advantages of SBRT over surgery is still lacking [18]. Given its potential for low-toxicity tumor ablation and its ability to be effective in combination with systemic treatments such as immunotherapy, SBRT is becoming a more widely used treatment [18]. It should be emphasized that SRS-SRT requires resources, training, and widely available equipment (e.g., magnetic resonance imaging) with accurate and reliable systems. Significant challenges in BM patients need to be highlighted, such as the wide range of tumor-patient characteristics, the nature of metastasis-directed therapies, and the integration of innovative and effective systemic treatment. In addition, individuals diagnosed with BM are often excluded from a significant proportion of clinical trials, leading to disparities and raising concerns about the applicability of the evidence specific to patients with intracranial extension [1,8,9]. To allow to conduct comprehensive comparisons, stratified BM patients must be eligible to participate in clinical trials. Patients with BM are a diverse group that has different primary tumors, treatment modalities, signs and symptoms, and life expectancies. Therefore, optimal management is a complex process that is influenced by several factors, such as performance status (PS), the type of cancer, the size quantity velocity of BM, and the availability of drugs that may effectively penetrate the central nervous system. The process of designing appropriate clinical trials for patients with BM remains a challenging undertaking. In 2023, a collaborative workshop organized by the National Cancer Institute highlighted the importance of establishing an agreement about reproducible and coordinated clinical investigation endpoints in the field of BM research [1]. 

This review provides a comprehensive overview of SRS-SRT as a treatment for BM, including various clinical considerations such as the accepted indications for SRS-SRT in patients with a limited number of BM, as well as the limitations of its approach in the case of multiple BM or in the case of BM located in the brainstem. In addition, the potential combination with systemic therapies, patient selection, as well as the health economic perspective of SRS-SRT will be explored. The aim of this review is to describe the potential benefits and limitations of stereotactic radiosurgery and stereotactic radiotherapy in the treatment of brain metastases.

## 2. Materials and Methods

A narrative literature review was conducted using the databases PubMed, Embase, Google Scholar, and Cochrane. The authors searched the databases until August 2023. Brain metastases, limited brain metastases, multiple brain metastases, brainstem metastases, radiosurgery, stereotactic fractionated radiotherapy, radiotherapy, consensus, expert recommendations, systemic therapy, immune checkpoint inhibitors, and immune radiotherapy were among the keyword combinations that were used. Following this, the results were filtered, and the authors examined therapeutic interventional studies, prospective and retrospective trials, that report on neurocognition, performance status (PS), quality of life (QoL), autonomy in daily activities, toxicity, intracranial progression-free survival (PFS), local control, distant brain control, neurocognitive performance, and PS preservation. Articles that were not relevant to the subject matter of our review were excluded. The authors prioritized prospective trials and meta-analyses to be described in the main text. A total of 993 articles were identified as matching our search terms. After applying filters, 137 documents were identified. Following a prioritization process, 95 were chosen for inclusion. 

## 3. SRS and SRT in the Management of a Limited Number of BM (1–4): An Accepted Treatment

In the early 2000s, the Radiation Therapy Oncology Group (RTOG)-9508 trial evaluated the SRS boost after WBRT in 333 patients with 1–3 BM [19]. This trial was designed to detect a 50% increase in median OS in the SRS boost group stratified by BM number (1 vs. 2–3) and RPA (I vs. II). Univariate analysis showed that the SRS boost group improved OS only for patients with a single BM (6.5 vs. 4.9 months, *p* = 0.039), and the multivariate analysis (MVA) confirmed this finding only for RPA class I patients (age 65, primary tumor controlled, KPS >60, no extracranial disease). Indeed, survival in BM patients is a complex outcome that is influenced by a number of factors, including but not limited to the primary tumor, age, PS, systemic disease status, and use of systemic treatment [20]. Sperduto et al. published a second retrospective analysis of the RTOG-9508 trial [21]. The authors stratified the patients according to a more precise prognostic indicator (GPA). When the entire cohort was analyzed, there was no difference in OS between the treatment arms (*p* = 0.78). However, regardless of the number of BM [1,2,3], the OS benefit of SRS boost was confirmed (21.0 vs. 10.3 months, *p* = 0.05) only for those in the best prognosis GPA group (score >3.5). In addition, the SRS group had significantly higher local control at 3 months (82 vs. 71%, *p* = 0.01). It should be noted that at this time, only 153 MRI sets (~60% of the patients) were available for central assessment, with 117 missing. Regarding neurocognitive outcomes, no difference in mental status was reported between the 2 arms at 6 months, based on the Mini-Mental Status Exam (MMSE). However, the MMSE is primarily used as a screening tool for dementia and is not designed to assess cognitive domains that are susceptible to impairment due to RT [22].

Due to concerns regarding neurocognitive deficits associated with WBRT, practices have evolved significantly over the past decade. Given the neurotoxicity of WBRT, the question has been whether SRS alone a sufficient option for patients with limited BM. Three randomized controlled clinical trials (RCTs) compared SRS with SRS + WBRT [13,14,15], and two RCTs compared local therapy (SRS or surgery) ± WBRT [23,24]. Patients with 1–4 BM less than 3–4 cm and an Eastern Cooperative Oncology Group (ECOG) 0–2 or Karnofsky performance status (KPS) of 70 were eligible. 

A meta-analysis of these trials was published in 2012, evaluating WBRT + SRS vs. WBRT (n = 2) and SRS vs. SRS + WBRT (n = 3) [20]. Local control was significantly improved with WBRT + SRS vs. WBRT (HR: 2.88, 95% CI 1.63–5.08, *p* = 0.0003; vs. HR: 2.61, 95% CI 1.68–4.06, *p* < 0.0001) [20]. 

WBRT also significantly improved distant brain control (HR: 2.15, 95% CI 1.55–2.99, *p* < 0.00001) [20]. It is important to note that the pooled hazard ratio for distant brain control was derived from three studies, namely Ayoma et al. [13], Chang et al. [15], and Kocher et al. [24]. The forest plot provided in the meta-analysis indicated homogeneity among the studies, as shown by a Chi-squared test *p*-value of 0.12 [20]. However, the I² value of 54% indicated a moderate to substantial level of heterogeneity. The study conducted by Kocher et al. carried more weight (60%) due to its large sample size (n = 359). Distant brain control is a precise indicator that includes the presence or absence of new BM. However, the definition of distant brain control lacked consistency and clarity, including a range of definitions such as the development of a new site [24] or a brain metastasis that is separate from the initial lesion treated with SRS, as observed on subsequent brain MRI [15] and not explicitly specified in one of the studies [13]. Due to the variable response criteria used in each study, the reliance on linear dimensions rather than volume measurements, and the lack of an approach to account for necrosis or pseudo-progression, the reliability of the outcome measure for local and distant control is compromised [25]. As a concrete example, Hong et al. explicitly defined distant intracranial failure as a new lesion appearing 1 cm or more from the baseline BM (see below) [23]. 

Furthermore, the local and distant control rates did not take into account the use of systemic treatment. Overall, despite the benefit in local and distant brain control, no benefit in OS was reported for the addition of WBRT compared to SRS alone [20]. Due to a lack of thorough assessments, the information on neurocognition, QoL, and toxicity was not addressed [20]. In fact, only one study used the Hopkins Verbal Learning Test (HVLT) as its primary outcome, and two out of five studies did not report it [20]. 

The neurocognitive outcomes of patients with BM undergoing SRS remain unclear and are of great concern to the community [1,26]. Neurocognition is a complex set of measures, and BM patients vary significantly in age, baseline function, tumor type, presence of extracranial disease, BM size, location, previous treatment (e.g., resection), response to treatment, and potential treatment-related side effects [26]. Currently, there is a lack of established screening tests to accurately identify patients most likely to experience delayed or permanent anatomic and functional brain toxicities [1]. Nevertheless, the authors concluded that SRS alone should be routinely presented to selected patients as a treatment option to be considered, along with frequent MRI-based follow-ups. 

Another thing to highlight is that most of these studies included patients mainly based on the number of BM with mixed tumor histology, predominantly patients with lung and breast cancer. In the context of the RTOG-9508 trial, Andrews et al. [19] stated that the necessity of WBRT combined with SRS boost for patients with radioresistant tumors remains an unresolved issue [19]. In 2009, Chang et al. suggested that omitting WBRT for 1–3 BM may be more appropriate for radioresistant BM (melanoma, renal cell carcinoma, and sarcoma), as WBRT may be less effective in this specific case [15]. Ten years later, Hong et al. reported a randomized phase III trial that included 215 patients with 1–3 melanoma BM [23]. The patients underwent local therapy in the form of SRS or surgery, after which they were randomly assigned to either WBRT or observation [23]. Approximately 60% of the patients had one BM, and more than half had an ECOG PS of 0. The volume of the BM ranged from 17 (5–47) to 18 (2–46) cc. Surgery was used as the primary local treatment in 60–64% of cases, SRS alone in 29–30%, and the combination only in 6–10% of cases. As a result, local failure at 12 months was significantly higher in the observation group (33.6 vs. 20%; OR: 0.49; 95% CI, 0.26–0.93). The results were derived from patients who received surgery alone without adjuvant SRS (42.2 vs. 20.3%; OR: 0.35; 95% CI, 0.16–0.77). Importantly, this difference in local failure was not present in patients with a single BM who received SRS (±WBRT) as local therapy (20.0 vs. 22.6%; OR: 0.86; 95% CI, 0.25–2.93). The results of the EORTC 29950–26001 trial provided additional evidence to support the aforementioned finding [24]. Indeed, surgery and WBRT had a higher rate of recurrence at the initial site after two years than SRS + WBRT. Surgery ± WBRT had 59 vs. 27% local recurrence, while SRS ± WBRT had 31 vs. 19% [24]. 

In the Hong et al. melanoma BM trial, adjuvant WBRT did not significantly improve clinical outcomes in terms of distant intracranial control (OR: 0.71;95% CI, 0.41–1.23), OS, neurological death, or PS preservation [23]. Interestingly, the lack of a significant reduction in intracranial failure differed from the meta-analysis by Tsao et al. (HR: 2.15, 95% CI 1.55–2.99) [20]. Despite the limitations of this HR value that have already been discussed, there are additional factors that may explain this discrepancy in the Hong et al. study. (i) The sample size of the study was overestimated to detect an absolute risk reduction of 22.0%, considering previous studies that included different histological cancer types, with lung and breast cancer being the most common and only ~10% being melanoma [11,12,13,26]. (ii) The radioresistance of melanoma and microscopic disease could not be effectively overcome by the WBRT dose of 30 Gy in 10 fractions. In addition, 24% of the participants received hippocampal-avoidance WBRT, which may have been the site of the recurrence, ∼9% according to the RTOG 0933 trial [27]. (iii) Overall, 77% had no systemic treatment at baseline, and immune checkpoint inhibitors were used only in less than 10% without detailed data provided. (iv) The omission of SRS on the surgical cavity resulted in significantly higher local failure. That could potentially contribute to the development of distant brain metastases. 

Indeed, a randomized phase III trial was conducted to compare complete surgical resection ± SRS in a cohort of 132 patients, the majority of whom (60–63%) presented with 1 BM [28]. More than 20–22% of the patients had a BM of melanoma. The 12-month local recurrence rate was 28 (vs. 57) percent in the SRS group (HR: 0.46; 95% CI 0.24–0.88; *p* = 0.015). The 12-month distant brain recurrence rate was not significantly different between groups; 33 vs. 42% (HR: 0.81; 95% CI 0.51–1.27; *p* = 0.35). The only significant determinant of distant brain recurrence was the presence of 1 vs. 3 BM at initial diagnosis (HR: 3.1; 95% CI 1.5–6.4; *p* = 0.0016). This finding may highlight the importance of tumor burden, considered as the amount of BM, and may potentially represent a degree of microscopic disease. This study did not provide data on systemic treatment characteristics. 

In addition, there has been increasing interest in the potential use of SRS as a definitive treatment option for small cell lung cancer (SCLC). The FIRE-SCLC cohort study was a comprehensive retrospective analysis conducted across 28 centers [29]. It included a total of 710 patients diagnosed with small cell lung cancer (SCLC) who had a majority of a limited (1–4) BM. In propensity score-matched analyses comparing SRS vs. WBRT, it was shown that WBRT was significantly associated with a better time to the central nervous system (CNS) progression (HR: 0.38; 95% IC: 0.26–0.55; *p* < 0.001). However, there was no significant improvement in OS or CNS progression-free survival. Given the possibility of further CNS progression and the need for additional BM salvage treatment, trials are underway to determine the value of using SRS alone versus WBRT for SCLC (Table 1).

Histology-specific investigations are essential in the examination of BM. However, conducting them can be challenging due to the presence of several potential factors that might influence the findings obtained from patients. Further study will be needed to determine the use and timing of SRS-SRT as a means of treatment for patients with multiple BM as the efficacy of molecular and immunotherapy growth. In addition, previously treated brain metastases should also be thoroughly investigated in specific trials.

Nevertheless, SRS alone is now recognized as the standard of care for patients with adequate performance status and 1–4 intact BM, and it is currently recommended by most international guidelines [2,30,31,32]. 

**Table 1 cancers-16-01093-t001:** Selected ongoing clinical trials assessing radiotherapy (SRS, SRT, and WBRT/HA-WBRT ± SIB) in patients with multiple brain metastases.

References	Setting Patients (n)	BM (n)	Main Inclusion/Exclusion Criteria	CITV	RT timing/ Systemic Agents Allowed	RT Scheme	Corticotherapy	Primary Endpoint	Statistical Endpoint	Completion Date
**HipSter** **(NCT04277403) *** **AUSTRIA** [33]	SRS vs. HA-WBRT + SIB N = 150 Phase III	4–15	KPS ≥ 70; PS ≤ 2 Exclusion: SCLC Brainstem metastasis Life expectancy < 3 months Any prior brain radiotherapy	25 cc		**SRS:** **80% isodose** 18–22 Gy; **SRT:** **80% isodose** 30 Gy-5 # **HA-WBRT** 30 Gy-12 # **SIB** 51 Gy-12 #		Intracranial PFS up to 18 months		February 2023
**CAR-Study B** **(NCT02953717)** **Netherlands** [34]	SRS vs. WBRT N = 81	11–20	KPS ≥ 70 Life expectancy > 3 months BM diagnosed on a triple dose gadolinium-enhanced MRI Exclusion: SCLC Lesion ≤ 3 mm from the optic apparatus Prior brain radiation Prior surgical BM resection	≤30 cc	Chemotherapy at the discretion of the physician	**SRS:** 18–25 Gy; **HA-WBRT** 20 Gy-5 # 30 Gy-10 #	Monitored and registered	Neurocognitive Performance: HVLT-R	HVLT-R: 5-point decrease from baseline based on 3 months	August 2023
**MDACC (NCT01592968) *** **USA** [35]	SRS vs. WBRT N = 88 Phase III Monocentric	4–≤15 or up to 20 at the time of treatment (once the head frame is in place)	Maximum diameter of largest lesion < 3.5 cm. Exclusion: Prior BM surgery Previous SRS (n = 1–3) delivered within 6 weeks SCLC Melanoma		Concurrent allowed at the discretion of the oncologist			Local control at 4 months Neurocognitive performance at 4 months: HVLT-R	HVLT-R 5-point decrease from baseline based on 4 months	September 2023
**Sunnybrook ** **(NCT03775330) *** **CANADA** [36]	SRS vs. SRS + (±HA-) WBRT N = 126 Parallel Prospective Observational Monocentric	5–30	KPS ≥ 70 HVLT-R ≥ 6 Exclusion: SCLC Previous SRS (n ≥ 5) delivered within 6 months		No concomitance allowed. Immunotherapy 1 week before/after Targeted therapies 1 day before/after Chemotherapy 1 week before/after	**SRS:** 15–20 Gy **SRT:** 24–27 Gy-3 # 25–32.5 Gy-5 # **(±HA-) WBRT:** 30 Gy-10 # 20 Gy-5 # +/−20% SRS reduction dose if SRS+ WBRT group		Neurocognitive performance: HVLT-R		December 2023
**CCTG CE.7 ** **(NCT03550391) *** **CANADA** [37]	SRS vs. HA-WBRT, Memantine Phase III Multicentric	5–15	PS ≤ 2 Lesion <2.5 cm Exclusion: SCLC Prior BM surgical resection Any prior brain radiotherapy BM located ≤ 5 mm optic chiasm/nerve. Use of NMDA agonists			**SRS:** 18–20 Gy 22 Gy **(HA-) WBRT:** 30 Gy-10 # + Memantine 20 mg		Overall Survival Neurocognitive progression-free survival		June 2024
**Dana-Farber ** **(NCT 03075072)** **USA** [38]	SRS vs. (HA-) WBRT N = 196	5–20	KPS ≥ 70 Exclusion: SCLC Lesion > 5 cm Age > 80 yo Any prior brain radiotherapy			**SRS-SRT:** 1–5 # **WBRT:** 30 Gy-10 # **HA** when possible		Quality of Life Survey at 6 months: MDASI-BT		July 2024
**ENCEPHALON-Trial** **(NCT03297788)** **GERMANY** [39]	SRS vs. WBRT, N = 56 Phase II Monocentric	1–10	ED SCLC Exclusion: KPS < 60 Any prior brain radiotherapy		Concurrent allowed, Last administration of Immunotherapy/targeted therapy/chemotherapy ≥ 1 week	**SRS:** **70% isodose** 20 Gy; < 2 cm 18 Gy; 2–3 cm **SRT:** **70% isodose** 30 Gy-6 #; <3 cm **WBRT:** 30 Gy-10 #		Neurocognitive Performance: HVLT-R	HVLT-R 5-point decrease from baseline based on 3 months	October 2024
**CYBERChallenge** **(NCT05378633) *** **GERMANY** [40]	SRS vs. WBRT N = 190 Phase II	4–15	NSCL Exclusion: SCLC >15 BM Any prior brain radiotherapy			**SRS** **WBRT**		Overall Survival Quality of life: EORTC QLQ-C15-PAL EORTC QLQ-BN-20		February 2025
**National Cancer Center ** **(NCT04452084)** **SINGAPORE** [41]	HA-WBRT vs. HA-WBRT + SIB N = 100	4–25 Phase II Monocentric	PS ≤ 2 Life expectancy > 6 months All histology Lesion or cavity < 5 cm Exclusion: Age > 80 Prior WBRT (prior SRS allowed) Concomitant systemic treatment	Total PTV < 60 cc	No concomitance allowed. Immunotherapy or Chemotherapy 7 days before/after Targeted therapies 3 days before/after	**HA-WBRT** 30 Gy-10 # **SIB** 40 Gy-10 # (surgical cavity) 45 Gy-10 #	Not mandatory but recommended if symptoms, edema, large target posterior fossa Memantine recommended.	Target lesion control: RANO-Criteria RECIST 1.1-Criteria	Target lesion control at 6 months	June 2025
**Dana-Farber ** **(NCT 03391362)** **USA** [42]	SRS Single Arm N = 100 Phase II	1–10	SCLC Exclusion: Lesion > 5 cm if not resected. Any prior brain radiotherapy			**SRS:** 20-1 # <2 cm 18-1 # 2–3 cm **SRT:** 30-5 # >3 cm		Death due to progressive neurologic disease		June 2025
**WHOBI-STER ** **(NCT04891471)** **ITALY** [43]	SRS vs. WBRT N = 100 Multicentric	≥5-unlimited ^1^	KPS ≥ 70 Life expectancy > 3 months Appropriate extracranial disease staging Controlled/ controllable extracranial disease. Exclusion: BM <5 mm from Hippocamp	Not including a maximum of BM, unless V12 Gy (1:10) V14 Gy (1:7)	Induction/Concomitant PD1-, PDL1-, CTL4-, BRAF- MEK-inhibitors	**SRS:** 15–24 Gy **SRT:** 27 Gy-3 # **WBRT:** 30 Gy-10 #	Dexamethasone 4 mg b.i.d 2 weeks	Neurocognitive performance, Moca Score HVLT-R Quality of life EORTC QLQ-C15-PAL BN-20 Autonomy in daily-life activities Barthel Index	30% between subjects of the two arms starting 6 months	September 2025
**NRG Oncology ** **(NCT04804644) *** **USA** [44]	SRS vs. HA-WBRT, memantine N = 200 Phase III Multicentric	≤10	KPS ≥70 De novo or recurrent SCLC Lesion ≤3 cm >5 mm from Hippocampi Exclusion: Any prior brain radiotherapy	30 cc	Initiation before RT allowed if symptomatic. Concurrent immunotherapy allowed. Concurrent chemotherapy not allowed.			Neurocognitive Performance: HVLT-R COWA TMT	Time to Neurocognitive Failure on at least on tests at 1 year HVLT-R COWA TMT	July 2030
**HIPPORAD ** **(DRKS00004598)** **GERMANY** [45]	HA-WBRT + SIB vs. WBRT + SIB N = 132	4–≤10 Phase II Multicenter Double blinded	Exclusion: SCLC Age > 80 yo BM < 7 mm from Hippocampi previous SRS-SRT (n > 1, >3 cm or n > 3, >1 cm) delivered within 3 months Previous Surgical resection within 4 weeks Brainstem metastases: N > 1, ≥5 mm, >2 cm		Administration of chemotherapy/immunotherapy/targeted therapy > 1 week before randomization	**HA-WBRT** 30 Gy-12 # **SIB** 42 Gy-12 # (surgical cavity) 51 Gy-12 #	Allowed	Neurocognitive Performance: VLMT	VLMT difference (word count, 0–75 words) at 3 months after radiation therapy and at baseline.	

SRS: stereotactic radiosurgery; SRT: stereotactic radiation therapy; WBRT: whole-brain radiation therapy; Gy: Gray; HA-WBRT: hippocampus-avoidance whole-brain radiation therapy, SIB: simultaneous integrated boost; NSCL: non-small cell lung cancer treatment; SCLC: small cell lung cancer; #: fractions; CITV: cumulative intracranial tumor volume; HVLT-R: Hopkins Verbal Learning Test – Revised; EORTC QLQ-C15-PAL: European Organization for Research and Treatment of Cancer Quality of Life Questionnaire Core 15 Palliative Care; EORTC QLQ-BN-20: European Organization for Research and Treatment of Cancer Quality of Life Questionnaire Core 20 Brain neoplasm; MRI: magnetic resonance imaging; yo: years-old; VLMT: Verbaler Lern- und Merkfähigkeitst est (German version of the Auditory Verbal Learning Test); PTV: Planning Target Volume; COWA: Controlled Oral Word Association; TMT: Trail Making Test (TMT) Parts A and B, ED: extensive disease; MDASI-BT/ MD Anderson Symptom Inventory-Brain Tumor; PFS: Progression Free Survival. ^1^: The WHOBI-STER study does not include a maximum number of brain metastases; V12 (1:10): the needed requirement is that the V12 of brain less PTVs should not exceed 10 times the number of metastases (1:10); V14 (1:7). *: protocol not published.

## 4. SRS and SRT in the Treatment of Multiple BM (> 4): A Matter of Debate

The most effective treatment for multiple brain metastases has become the subject of increased debate in the past decade and is an investigation of significant epidemiologic relevance. Several advanced technologies, such as patient setup, target localization, treatment planning, and delivery, have changed and personalized the way radiation is delivered to BM patients [6]. Although there are not enough data from trials to show that WBRT is a better option than SRS alone for patients with multiple BM, it has long been accepted practice that these patients should receive WBRT.

In surgical trials, "limited" usually means a patient with only one BM, and the term has changed since the introduction of SRS to include patients with up to four BM [46]. This threshold remains to be explored, and a one-size-fits-all approach is not appropriate, as various factors influence outcomes in BM. Due to the lack of published RCTs, SRS in patients with >4 BM is not well established and is considered highly controversial [2,26]. One of the first prospective randomized multicenter phase III trials in 4–10 BM comparing WBRT with SRS for QoL, OS, and brain failure-free survival was prematurely terminated due to insufficient accrual [47]. Interestingly, the low accrual (29/230) was due to the patient or referring physician’s preference for SRS over WBRT despite the lack of level I evidence. 

RCTs are currently underway in patients with 5–15 BM (Table 1). The most effective way for physicians to treat patients with multiple BM will be debated until these trials are published. Nevertheless, in recent years, SRS has become increasingly common in patients with multiple BM. 

The JLGK0901 prospective observational study, which included 1194 fit patients with 1–10 BM who have been treated with SRS alone, demonstrated no inferiority in OS between those with 2–4 vs. 5–10 BM, with the caveat of a total cumulative intracranial tumor volume ranging from 0.02 to 13.9 cc (≤15 cc) [48]. There was also no difference in neurological mortality (6–10%), neurocognitive function, local recurrence, new lesion, or salvage treatments (WBRT, surgery, and SRS) [48]. These results have been validated by long-term evaluations (e.g., local control, Mini-Mental State Examination, and treatment-related complications) [49]. The controlled status of extracerebral disease prior to SRS was found to significantly favor longer survival (HR: 0.27; 95% CI 1.101–1.469; *p* = 0.0011) compared to cases in which the extracerebral disease was not controlled [48]. It is also worth noting that the majority (92%) of deaths were attributed to the progression of the systemic disease, underscoring the continued importance of disease progression as the primary factor leading to death [48]. It should be noted that in the JLGK0901 study, only 17% of the cohort had 5–10 BM, and the median number was 6 BM.

Recently, at least three large retrospective studies evaluating SRS in patients with 5–15 BM have been reported [50,51,52]. In 2017, Ali et al. evaluated 5750 patients treated with SRS alone and found significant but mild OS differences for 1 vs. 2–10 (HR = 1.110) and 2–10 vs. >10 (HR = 1.128) [50]. After controlling for age, KPS score, systemic disease state, and cumulative tumor volume, each 6–7 BM (excluding breast cancer) increased mortality by ~4%. This highlights that the number of BM is an inadequate predictor of survival; therefore, the choice of therapy should be based on additional factors. In fact, cumulative intracranial tumor volume (CITV) was significantly associated with a survival HR of 1.015 per cc [50]. This means that the evaluation of BM should not only focus on its amount but also consider CITV. 

Hughes et al. published the results of a multi-institutional study that included 2089 patients who underwent SRS alone for 1, 2–4, and 5–15 (n = 212) BM [52]. No difference in survival was found between 2 and 4 vs. 5 and 15 BM after univariate and multivariate analysis. However, the 1-year distant brain failure rate in the 5–15 BM groups was 50%, which was significantly higher than the 41% rate in the 2–4 BM groups. The cumulative number of new BM that had developed since the initial SRS was also significantly higher in the 5–15 group (11.7; IQR, 4.6–29.8) compared to the 2–4 BM (6.1; IQR, 2.4–16.1). These findings suggest that the amount of BM at the time of diagnosis may be associated with the potential of BM to spread to the brain. It is important to remember that the JLGK0901 study did not provide such results. In fact, patients with 2–4 vs. 1 BM experienced more new lesions (HR: 0.55; 95% CI, 0.46–0.66; *p* < 0.0001) and repeat SRS (HR: 0.57;95% CI 0.46–0.71; *p* < 0.0001), but these outcomes were not significantly different between the 2–4 vs. 5–10 groups [48]. The results of JLGK0901 may have been influenced by the limited sample size of patients with 5–15 BM (17%, 208/1194) and the fact that 10% of patients had no MRI data [48]. In order to draw meaningful conclusions within the group of patients with more than 5 BM, it is important to ensure that this subgroup is adequately represented. 

In this context, a single institution SRS result for 2–4 vs. 5–15 (representing ~ 43% of the cohort) BM based on 2193 patients was retrospectively reported by Yamamoto et al. [51]. When tumors from SCLC, breast, and kidney cancer were excluded, the median OS was significantly longer in the 2–4 BM group (8.1; 95% CI; 7.4–8.8 months) compared to the 5–15 BM cohort (7.2; 95% CI; 6.6–7.8 months); HR: 1.169; 95% CI: 1.065–1.283; *p* = 0.0010. Multivariate analysis identified female gender, a KPS score ≥80, a diagnosis of NSCLC (as opposed to gastrointestinal cancer), controlled initial disease, and the absence of extra-cerebral metastases in both groups as predictors of longer survival. In addition, there were no significant differences between the two groups in the cumulative occurrence of neurological death, salvage SRS, or SRS-related comorbidities. The cumulative events of neurological disability (HR: 0.741, 95% CI: 0.569–0.966, *p* = 0.0026) or local recurrence (HR: 0.626, 95% CI: 0.405–0.970, *p* = 0.0035) were significantly lower in the 5–15 BM group. These results could be explained by the fact that the neurological deterioration (9.4 vs. 12.6%) was characterized by an unreliable endpoint of "20% decrease in KPS score from baseline", which could also be attributed to distant progression.

Although the 2–4 BM group had a significantly higher median peripheral dose (24 vs. 22 Gy) and a larger median tumor volume (3.98 vs. 3.40 cc), they had significantly more local recurrences (4.6 vs. 7.3%) compared to the 5–15 BM cohort. It is important to note that a significant proportion (30%) of individuals in both groups did not have accessible MRI data, which may have contributed to this unexpected finding. 

Salvage WBRT was required by significantly more patients in the 5–15 tumor group than in the 2–4 tumor group (HR: 2.165, 95% CI: 1.233–3.803, *p* =.0072) 3.5 vs. 1.7%, respectively. In contrast, in the multi-institutional study by Huges et al., WBRT was administered at the discretion of the treating physician, so institutional bias (such as providing the best supportive care at progression in the 5–15 BM group) may have caused the lack of a significant difference in the time needed to salvage WBRT (2–4 vs. 5–15 BM; ~5 months) [52]. This salvage treatment emphasizes the importance of close MRI follow-up when treating with SRS alone, as 50% of patients will develop new BM, and distant brain failure increases with time [46].

The retrospective nature of these studies should be taken into account when interpreting them, as well as (i) the long inclusion period (e.g., 1991 to 2018), (ii) the heterogeneity of the primary tumor type (mainly lung and breast cancer), (iii) the paucity of information regarding the molecular tumor characteristics and systemic treatment, (iv) the inclusion of individuals who have already undergone brain therapy (e.g., WBRT, SRS, and surgery), (v) the effect of inter-institutional bias on patient selection, (vi) the lack of details regarding the extracerebral status, (vii) the limited follow-up due to poor OS, and (viii) the lack of neurocognitive and QoL data especially at the baseline (50–52). 

However, these are hypotheses generated to clarify the most effective way to use SRS alone in patients with five or more BM, a treatment that appears appropriate in carefully selected patients. According to the voting of the American Radium Society panelists (radiation oncologists, neuro-oncologists, and neurosurgeons), SRS alone was appropriate for patients with 0.5 cm of asymptomatic BM from NSCLC (without targetable mutations) with 2–4 or 5–10 BM at initial diagnosis (case 1) or extracranial progression on systemic treatment (case 2) [26]. SRS alone in case 2 was appropriate for 11–15 BM with a disagreement in case 1, and the opposite was true for 16–20 BM (case 2: disagreement; case 1: agreement). SRS alone was inappropriate for >20 BM in both scenarios. Several panelists recommended SRS for patients with >20 BM, although all agreed that further investigation is needed. Finally, there was widespread agreement about appropriate supportive care for NSCLC patients with KPS 60 and 6 asymptomatic BM.

It is important to highlight that despite the fact that WBRT is based on the concept of palliation, the QUARTZ trials found no difference (e.g., overall survival, quality of life, and dexamethasone use) between WBRT (20 Gray in 5 fractions) and optimal supportive care for patients with poor prognosis non-small-cell lung cancer (NSCLC) with asymptomatic BM and ineligible for surgery or SRS [53]. In this study, neurosurgeons and radiation oncologists assessed patients as “unfit for surgical resection or SRS”, although the actual criteria used in each case were not reported. It is important to remember that the study did not report neurologic mortality but rather presented measures of overall survival (OS) and quality of life (QoL). Therefore, the omission of the WBRT should not be applied to all NSCLC patients. A one-size-fits-all approach is no longer appropriate. 

Nowadays, WBRT is no longer the traditional approach for BM patients as the treatment paradigm is rapidly changing and advancing [12]. The rapid pace of technical advancements in RT has outpaced the availability of clinical evidence. Consequently, there is an urgent need for high-quality level 1 evidence to provide a more accurate understanding of the role of SRS in patients with multiple BM. The factors influencing the inferiority or non-inferiority of SRS compared to WBRT should be carefully investigated. There are RCTs (NCT04891471, NCT02953717, NCT01592968, NCT03775330, NCT03550391, NCT 03075072; etc.) that are underway to compare SRS with WBRT in patients with multiple BM. However, an important discrepancy has been noted in these trials, as summarized in Table 1. Nevertheless, if SRS-SRT is chosen as a treatment option for multiple BM, it is essential to emphasize the importance of close follow-up with sequential MRI (at least once every two to three months during the first year after treatment or whenever there is a possible neurological progression). In fact, SRS-SRT does not have a prophylactic purpose compared to WBRT, and close surveillance is necessary for the early diagnosis of new BM. The SRS-SRT technique effectively delivers a highly targeted and heterogeneous dose specifically to the metastatic site while ensuring tiny exposure of a healthy brain and a low mean brain dose. In contrast to WBRT, this treatment can be repeated as needed and is more convenient for patients, as SRS is usually delivered in only 1 to 5 sessions (compared to 10 fractions in the case of WBRT). A clinical case of SRT management for multiple BM is presented in Figure 1 and illustrates subsequent SRT treatments in a patient with an NSCLC and 16 BM at first SRT.

## 5. Brainstem Lesions: Reaching the Limits of SRT

The percentage of secondary intracranial tumors that metastasize to the brainstem is estimated to be 3–7% [54,55]. Metastatic tumors located in the brainstem pose a unique challenge for therapeutic decision making. Brainstem metastases (BSM) are associated with acute and severe neurological deterioration and are rarely removed surgically due to the inherent risks associated with their anatomic location [56]. Compared to supratentorial metastases, BSM is associated with an increased likelihood of death (HR: 3.52; 95%CI, 1.81–6.85%) and appears to be an independent indicator of inferior survival [57]. Patients with BSM were excluded from nearly all landmark randomized SRS trials, data are limited to retrospective series, and the efficacy and safety of SRS are not well reported [58,59]. Furthermore, in contrast to the reported cohort of BSM patients, BM trials had a higher percentage of solitary mets, controlled extracranial disease, and patients with favorable performance status [59]. Patients with BSM were included in the JLGK0901 trial, but their results were not reported separately [48]. As a result, BSM radiation techniques, dose, and fractionation lack evidence-based standards, and the clinical management of these patients remains controversial. Concerns regarding the use of SRS as a therapeutic approach are generally attributed, in part, to previous studies that suggested a maximum radiation dose (12–12.5 Gray) to the brainstem when administered in a single fraction [60,61]. These studies included a limited patient cohort of patients with a variety of targets, including both benign and malignant cases, and involved a wide range of volumes, prescriptions, and SRS techniques, such as Gamma Knife and frameless linac-based approaches. Nevertheless, in 2010, the Quantitative Analyses of Normal Tissue Effects in the Clinic (QUANTEC) study found that exceeding the maximum of the brainstem dose (per single fraction) of 12.5 Gy could result in an increased likelihood (>5%) of adverse outcomes, such as permanent cranial neuropathy or necrosis [62]. However, it should be emphasized that they included only acoustic tumors to define this threshold [62]. BSM are unlikely to respond efficiently to 12.5 Gy (BED (α/β_10_) = 28 Gray), and higher SRS doses (e.g., >16 Gy) have been widely reported retrospectively for BSM [56,58,59,63]. The report published by the AAPM Task Group 101 stated that 0.5 cc of the brainstem has the capacity to tolerate a maximum dose of 15 Gy in a single fraction [64]. 

In fact, the study of radiotherapy-induced brainstem injury is difficult due to several factors. The incidence of such an injury is usually rare and tends to manifest several months to years after treatment. Patients with intracranial tumors, especially those with shorter survival periods, often face challenges in distinguishing between adverse effects and disease progression [60]. In addition, the assessment of injury induced by clinical radiotherapy presents a challenge in terms of inter-physician reliability [63]. 

The use of SRS has typically been limited by concern about potential adverse effects, but over the past twenty years, an increasing number of studies have demonstrated that SRS is effective in the treatment of BSM. In 1993, Somaza et al. retrospectively reported the first cases of BSM from melanoma treated with SRS combined with WBRT [65]. In one patient, they achieved 10 months of local control using Gamma Knife (16 Gray to the 50% isodose line) with WBRT (30 Gray in 12 fractions), while the second patient experienced grade V intracranial hemorrhage 7 months after SRS [65].

In 2016, the International Gamma Knife Research Foundation (IGKRF) published the results of a multi-institutional retrospective study of 547 patients (596 BSM) [56]. They reported a local control rate of 81.8% at 1 year and an overall grade 3–4 toxicity rate of 7.4%, with no grade V. The primary histologies were NSCLC and breast cancer; 49% had previously received WBRT; the median BSM volume was 0.8 (0.01–21) mL; and they were treated with a median margin dose of 16 (8–25) Gray. According to their findings, three variables—age (≥65), a margin dose of <16 Gy (vs. ≥20 Gy), and a maximum dose—were associated with a significantly increased risk of local failure. It was also shown that tumor volume, margin dose, and recent (less than 4.5 months) WBRT increased the risk of severe toxicity (OR:4.7; 95% CI: 1.8–11.4). They found that the tumor histology had no effect on the rate of local failure, nor did they report any effect on the location of the BSM or the volume of the brainstem that had received 12 Gy. 

Chen et al. conducted a meta-analysis in 2021 that evaluated the efficacy and safety of SRS in a population of 1,446 patients (1590 BSM) across 32 retrospective studies (excluding the 547 patients from the IGKRF cohort) [59]. The study population was diverse, with a preponderance of NSCL and breast cancer. The median volume of BSM was 0.40 (0.0025–24.88) cm³. The majority of the cohort (74%) underwent Gamma Knife treatment, with a median prescribed dose of 16 (6–30) Gray. The authors reported an 86% (95%CI, 83–88%; I^2^: 38%) rate of local control at one year, accompanied by a 2.4% (95%CI, 1.5–3.7%; I^2^: 33%) incidence of grade ≥ 3 toxicity. A significant proportion of treatment complications were observed in patients diagnosed with melanoma or renal cell carcinoma, those who had received or were receiving WBRT, and those with a larger BSM volume (median of 1.7 cm³). In such circumstances, the possibility of dose reduction or SRT may be considered [7]. 

Nowadays, there is a growing trend to replace headframe-based SRS with frameless image-guided SRS, which offers improved patient comfort and scheduling flexibility [7]. In this context, Nicosia et al. performed a multicenter retrospective analysis of the efficacy and side effects of frameless linac-based SRS-SRT in 105 patients (111 BSM) [58]. They included patients with 1–2 BSM; 73% had concomitant systemic therapies (e.g., chemotherapy, targeted therapies, immunotherapy), 14% had prior WBRT, 52% of lesions were ≤10 mm and the median BSM volume was 0.4 (0.02–23.6) cc. In this study, they used SRS for ≤10 mm BSM and SRT for larger lesions. The 1-year freedom from local progression (FLP) was 90.4%. No 90-day severe acute toxicity was reported; local BSM progression caused 2.8% grade V toxicity. SRS and SRT have similar FLP, especially when BED (α/β_10_) >35 Gray. The multivariate analysis of the study revealed that two covariates, namely BSM ≤ 0.4 cc and BSM located in the pons, showed a significant increase in FLP. Prior WBRT was an independent variable associated with worse cancer-specific survival and higher neurological mortality. Interestingly, concurrent targeted therapy (e.g., anti-HER2, ALK inhibitor, tyrosine kinase inhibitor, VRAF inhibitor) was significantly associated with improved OS and cancer-specific survival.

Although the brainstem represents only around 2% of the total brain, it contains an abundance of tightly interconnected, vital neuronal pathways that are essential for sustaining life [66]. Since almost all randomized SRS trials exclude patients with BSM, an effective and safe therapy such as SRS-SRT may be a crucial steppingstone to allow enrolment in clinical trials. This lack of evidence-based requirements for BSM must be handled in the hands of experienced clinicians. To create an effective and robust BSM complication probability model, future SRS-SRT research should collect and publish significant dosimetric parameters. In patients with BSM, the choice of tumor margin dose and SRS techniques can have a significant influence on the rate of off-target dose decay and the degree of radiation-related neurological toxicity [56]. In addition, the radiation oncologist must use recent (e.g., ≤seven days from the treatment) 3-dimensional magnetic resonance images with ≤1.5 mm slice thickness to accurately delineate the brainstem and gross tumor volume (GTV), with particular emphasis on the boundaries and planification margins should be reduced as much as possible or even to zero. Knowing that the superior extension and cerebral peduncles can sometimes be indistinct, and that tumor cells or surgery can change the position of the brainstem. In addition, the investigation of the safety and efficacy of concurrent immunotherapy and targeted therapies in BSM with SRS remains poorly investigated/understood [56,58,59]. Figure 2 shows an example of a BSM arising from melanoma, which was treated with a single session of SRS at a dose of 24 Gy at 70% isodose without margin.

## 6. The Combination of SRS-SRT with Systemic Treatments

The exclusion of patients with BM is common in randomized trials of new systemic therapeutic strategies, leading to a lack of knowledge about BM. Standard systemic chemotherapeutic agents have limited efficacy in the treatment of BM due to the efflux pumps and blood–brain barrier and the differentiated tumor microenvironment [67,68]. This limited penetration supports the clinical difference between intra and extracranial responses. However, due to recent advances in systemic therapy and molecular mutation identification, patients can now benefit from systemic drugs (e.g., targeted and immunotherapy) based on their cancer type and genetic and molecular markers [5]. 

In 2022, the American Society for Clinical Oncology/Society for Neuro-Oncology/American Society for Radiation Oncology (ASCO-SNO-ASTRO) published comprehensive guidelines to direct clinicians in the treatment of patients with BM from solid tumors [9]. According to these guidelines, patients with symptomatic BM should receive local therapy regardless of systemic treatment. However, patients with asymptomatic BM could delay local treatment until intracranial progression is manifest. This recommendation applies to NSCLC with EGFR mutations or ALK rearrangements, BRAF-V600E melanoma, or human epidermal growth factor receptor 2-positive metastatic breast cancer treated with targeted therapy (e.g., asimertinib/icotinib; alectinib/brigatinib/ceritinib; tucatinib/trastuzumab) or immune checkpoint inhibitors (ipilimumab-nivolumab). Note that the quality of the evidence is low, and the recommendation is weak. Interdisciplinary discussion is essential when considering deferring local treatment, and we can emphasize that from a dosimetric point of view, treating low-volume BM results in lower doses of healthy brain tissue.

BM has a high degree of heterogeneity, and the potential correlation between specific genetic characteristics together with different immunological states remains to be explored [69].

It is important to emphasize that genetic divergence between BM and the primary neoplasm may result in decreased response to targeted agents. In this context, oncogenic alterations in BM may explain the discrepancy between intra- and extracranial responses. Brastianos et al. reported 86 whole-exome sequencing cases of patients with matched normal tissue, primary malignancies (lung: 44%; breast: 24%), and BM. In all samples, BM and primary tumors were found to have evolved separately from a common ancestor, and interestingly, 53% of BM harbored novel clinically actionable mutations [70]. Next-generation sequencing (NGS) has revealed that genomic alterations allow cells to migrate to the brain, and the genetic events that cause tumorigenesis in BM are still under investigation [5]. 

An important area of investigation in neuro-oncology is the optimal timing of effective systemic treatments with respect to SRS-SRT. Currently, there is interest in the potential synergistic effects of immunotherapy combined with SRS for the treatment of brain disease, with several trials underway. Indeed, RT has the ability to produce a number of "danger signals", including the production of neoantigens via the death of tumor cells that may trigger a response [71]. It is important to note that only a tiny amount of neoantigen can activate an immune response, and its production alone is not sufficient [71]. Nevertheless, the brain tumor microenvironment (TME) is a dynamic and complexly interconnected system that remains poorly understood.

Several retrospective studies and an individual patient data meta-analysis have shown that concurrent SRS + immune checkpoint inhibitors (ICIs) improve BM outcomes compared to non-concurrent regimens [72,73,74,75,76,77,78]. However, it is interesting how broadly each study defines "concurrent" (e.g., ≤1 week; within 4 weeks; ≤30 days; with an interval ≤5 half-lives of the ICI) [73,74,75,76,77]. One of these studies found that ICI-exposed lesions (n = 196) had a lower lesion reduction rate than ICI-naive lesions (n = 553) (45% vs. 63%), with the highest responder among ICI-naive lesions that underwent "immediate" SRS, defined as 1 biological half-life of the ICIs [79]. 

Conflicting results on the adverse events showed that SRS and ICIs did not increase the rate [79,80], while other results found an increase [81,82]. This may be due to the different follow-up times in the studies, which require larger sample sizes and longer follow-ups. Lehrer et al. published the results of 650 patients with 4182 NSCLC, renal cell carcinoma, and melanoma BM treated with SRS ± ICIs. They found that the rate of symptomatic radionecrosis (RN) between 1 and 2 years was 4.8% and 7.2%, with a median follow-up of 12.8 and 14.1 months for the concurrent and nonconcurrent groups, respectively [83]. Interestingly, V12 Gy (<12 vs. 12–20 vs. >20 cm^3^) was interestingly confirmed as a significant variable predictor of RN. In this study, concurrent administration of ICIs (within 4 weeks) did not increase the risk of RN.

Clinical trials are underway to determine the optimal sequence between SRS-SRT and ICIs. For example, the addition of SRS (16–22 Gray) or SRT (24–30 Gray) in combination (within 7 days) with ICIs (nivolumab and ipilimumab) is being evaluated in the randomized phase II ABC-X trial (NCT03340129) [84]. With a planned completion date of 2025, the trial will enroll 218 asymptomatic, treatment-naive (≥1 BM, ≤40 mm) patients with BM of melanoma. The primary endpoint is the neurologic-specific cause of death at 12 months [84]. 

## 7. Clinical Considerations for Patient Selection and Eligibility

According to data from 1987, the median survival duration for patients diagnosed with BM who were treated with steroids alone was ~2 months, while patients who received no therapy had a median survival duration of ~1 month [85]. The primary goal in determining the most effective therapy for a given patient is to optimize either the QoL or OS [7]. Due to the wide range of tumor-patient-related characteristics and evidence supporting different therapeutic approaches, the management of BM is challenging difficult, and should be addressed multidisciplinary [5]. In fact, it is necessary to accurately assess the prognosis following BM treatment in order to guide patient-clinician decision-making. Furthermore, patients with BM are typically ineligible for clinical trials on the outdated basis of their low survival rates, which may obscure the benefit of the treatment under investigation [8]. These concerns have led to the establishment of several prognostic classification systems to select patients and provide realistic survival predictions, including the Recursive partitioning analysis (RPA) [86], the Score Index for Radiosurgery (SIR) [87], the Basic Score for Brain Metastases (BS-BM) [88], the Graded Prognostic Assessment (GPA) [89], and the Disease-Specific (DS)-GPA [90]. 

Newer prognostic parameters, such as cancer genetic, molecular, and histological alterations, are now included in the recently updated DS-GPA score [8]. However, some limitations should be kept in mind: (i) retrospective design, (ii) predominance of breast and NSCLC (55% of the cohort), (iii) 60–81% of patients had limited (1–3) BM, (iv) 56–85%, had extracranial metastases without further precision (e.g., number, location, systemic therapies), (v) recurrent BM and/or leptomeningeal carcinomatosis patients were excluded, (vi).

Given the diversity of BM patients, prognostic classification systems are useful for patients and multidisciplinary teams. In fact, they could provide specific survival projections, as well as education, end-of-life decisions, assessing the advantages/risks of different therapeutic approaches (aggressive vs. palliative), and provide comprehensive data on BM patient stratification [5,6]. The importance of primary tumor heterogeneity in BM patients necessitates the development of improved prognostic classification methods that reliably predict survival rates, which would be of critical value to both patients and physicians [5].

## 8. Health Economic Perspective

Due to limitations in the epidemiologic data, it is difficult to estimate the true population-based incidence of BM and the rate of neurological death by histologic or molecular subtype [1]. Consequently, it may be a significant difficulty to identify patients at higher risk of developing BM and to develop a standard MRI procedure based on each cancer subtype.

Retrospective comparisons of the presentation and management of BM patterns in patients with lung cancer (n = 659; screened group with brain MRI) and breast cancer (n = 349; not screened) [91]. Significantly more were reported: symptoms, numerous (>4 BM) larger and brainstem-located BM, leptomeningeal disease, and WBRT as initial treatment in the unscreened group. In addition, the unscreened group had a higher rate of neurological death (37.3% vs. 19.9%; *p* < 0.001) and time-to-event-based outcome (HR: 1.54; 95% CI, 1.10–2.17; *p* = 0.01). Significantly, 24% of the unscreened group were screened. This screening was performed either at the patient’s request or as part of their enrollment in a clinical trial, which reflects real-world clinical situations. However, a screening approach for all cancer patients appears to be economically and technically unsustainable.

In the current scenario of escalating healthcare expenditures, there has been a significant increase in international attention to the economic evaluation of SRS as a BM treatment option [92,93]. The challenge of defining the usefulness of comparative cost-effectiveness studies is compounded by the complexity arising from variations in cost definitions and heterogeneity in healthcare reimbursement. Nevertheless, recent health economics studies have reported that SRS is a more cost-effective option than WBRT for patients with up to three BM with a good performance status and a longer expected survival. In addition, SRS has also been found to be a better alternative to surgery for lesions with limited mass effect [92,93]. However, in situations where the PS is poor or the predicted survival rate is low, WBRT has been found to be the more economically feasible alternative [93]. It is important to remember that the SRS-SRT process should not be viewed as a single event but rather as a complex set of processes, including the allocation of advanced technological resources with the goal of providing targeted and tailored therapy. Finally, involving patients in the decision-making process for treatment selection is critical to fully understanding their priorities and concerns. Factors such as quality of life, the ability to maintain functional independence, and the impact of treatment on survival are highly relevant in influencing their decision [94].

So far, evidence for SRS alone in patients with BM has come from studies in which patients were mostly included and accepted based on the BM number. Even though the amount of BM has been shown to be important, it is not the only one. Factors such as the location (e.g., the brainstem), the CITV, the kinetic, and the systemic treatment should also influence outcomes but have not been significantly investigated. In addition, previously treated brain metastases should also be thoroughly investigated in specific trials.

The variability observed in this context has resulted in a lack of consistency when comparing the effects of treatment across various studies. As an example, the Response Assessment in Neuro-Oncology Brain Metastases (RANO-BM) criteria should be used instead of the RECIST criteria to accurately evaluate both local and distant recurrence [25]. Importantly, mapping the recurrence’s location with the radiotherapy isodose must be mandatory and provided. A range and combination of cognitive function monitoring assessments, functional status, toxicity, health-related QoL, and patient-reported outcomes (PROMs) that are specific to the side effects of radiation should all be included in the study’s results. The Brain Symptom Impact Questionnaire (BASIQ) and the Functional Assessment of Cancer Therapy General and Brain (FACT-G; FACT-B) were the sole two PROMs that were explicitly established in the BM patient [95]. A complete understanding of the long-term evolution of patients with BM is necessary. An area requiring effort is the development of instruments developed specifically for BM patients that may be easily implemented by multiple organizations into everyday clinical practice [1].

## 9. Conclusions

Modern BM management is a major challenge in cancer treatment. During the last decade, the therapeutic paradigms for the majority of BM patients have changed significantly to an active strategy. Improvements in early detection and the development of more effective systemic treatments have significantly increased the life expectancy of patients with brain metastases. Therefore, maintaining functional independence while controlling intracranial disease has become critical. In addition, a comprehensive understanding of the intercellular interactions that occur within the cancer microenvironment is paramount to the advancement of innovative trials. SRS-SRT is a well-established therapeutic modality that is now undergoing an important evolution in its overall paradigm. The phenomenon of change presents several unanticipated opportunities and challenges that require further investigation. A one-size-fits-all therapeutic approach for all patients is no longer true.

## Figures and Tables

**Figure 1 cancers-16-01093-f001:**
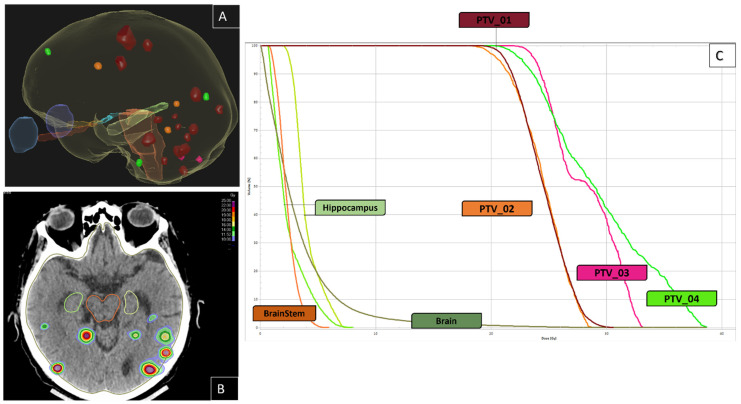
Illustration of a case of multiple brain metastases from lung cancer without drivers’ mutation treated in August 2018. In February 2021, he developed 16 BM. He had four sessions of SRS (20 Gray to the 70% isodose line): PTV_01 in February 2021 (n = 16); PTV_02 in May 2021 (n = 3); PTV_03 in August 2021 (n = 2); and PTV_04 in October 2021 (n = 3). He had no new brain metastases until the last MRI in April 2022. (**A**): three-dimensional image of the brain and the target BM: red: PTV_01; orange: PTV_02; pink: PTV_03; green: PTV_04. (**B**): axial CT-scan slice, showing the patient’s dose-color-wash, (**C**): Dose-Volume Histogram.

**Figure 2 cancers-16-01093-f002:**
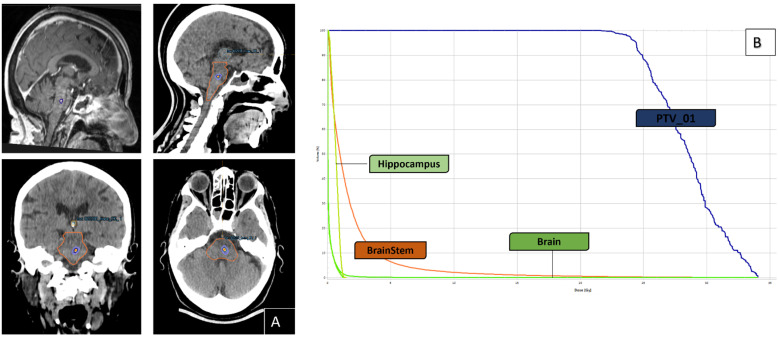
Illustration of the case of a single BSM from a melanoma without drivers’ mutation treated with one course of SRS (24 Gray to the 70% isodose line). (**A**): axial–sagital CT-scan and MRI slice, showing the target lesion and the patient’s dose-color-wash; (**B**): Dose-Volume Histogram.

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
