# Peer review of "Stereotactic Radiosurgery and Stereotactic Fractionated Radiotherapy in the Management of Brain Metastases"

_cancers, 2024, doi:10.3390/cancers16061093_

Round 1
Reviewer 1 Report
Comments and Suggestions for Authors
This is a well-written review on the management of intracranial metastases using stereotactic radiosurgery (SRS) and stereotactic fractionated radiotherapy (SFR). I suggest using either 'brain' or 'intracranial' but not both in the title for conciseness.
The widespread adoption of targeted therapy and immunotherapy has led to a significant improvement in the treatment of systemic malignancies. Both SRS and SFR play crucial roles alongside these treatment options, markedly enhancing patient survival. It would be beneficial to prioritize studies examining the combination of SRS-SFR with targeted therapy and/or immunotherapy for brain metastases.
Additionally, clearly stating the class of evidence would enable readers to better grasp the findings of the studies and apply them in clinical practice.
In line 74, 'SBRT' should be spelled out as 'stereotactic body radiotherapy.' Additionally, 'SRS' was spelled out twice, in line 57 and 61, which can be streamlined for clarity. The phrase 'stereotactic radiosurgery' was redundant in line 100.
The statement 'a formal literature search was not performed' may diminish the scientific rigor and persuasiveness of this review.
Comments on the Quality of English Language
Author Response
Reviewer #1:
- This is a well-written review on the management of intracranial metastases using stereotactic radiosurgery (SRS) and stereotactic fractionated radiotherapy (SFR). I suggest using either 'brain' or 'intracranial' but not in the title for conciseness.
Response:
The authors acknowledge the reviewers' comments. We changed the title by “Stereotactic radiosurgery (SRS) and Stereotactic fractionated radiotherapy (SFR) in the management of brain metastases.”
- The widespread adoption of targeted therapy and immunotherapy has led to a significant improvement in the treatment of systemic malignancies. Both SRS and SFR play crucial roles alongside these treatment options, markedly enhancing patient survival. It would be beneficial to prioritize studies examining the combination of SRS-SFR with targeted therapy and/or immunotherapy for brain metastases. Additionally, clearly stating the class of evidence would enable readers to better grasp the findings of the studies and apply them in clinical practice.
Response:
The reviewers' thoughtful comments are much appreciated by the authors. The combination of SRS-SRT with targeted therapy and/or immunotherapy for brain metastases is covered in detail in the authors' chapter, "The combination of SRS-SRT with systemic treatments." On top of that, authors made comment of the combination in every section of the review. There is currently insufficient reliable data to make any clear recommendations. For instance, in the Hong et al. melanoma BM study, immune checkpoint inhibitors were utilized in less than 10% of cases, without detailed information available, and 77% of patients did not receive systemic treatment at baseline (lines 216-218). As a second example, data on the characteristics of systemic treatment were not provided in the Mahajan et al. study (line 230). Once more, we emphasized in the chapter how little is known about "the molecular tumor characteristics and systemic treatment" (line355). The American Society for Clinical Oncology/Society for Neuro-Oncology/American Society for Radiation Oncology (ASCO-SNO-ASTRO) comprehensive 2022 was thoroughly addressed by the authors. Furthermore, we provide a column in Table 1 labeled "RT timing/systemic agents allowed" for the systemic treatment.
Concerning the evidence class, the authors consistently specified the study type (retrospective, prospective cohort, RCT, meta-analysis) and provided crucial information (design, inclusion criteria, population characteristics). Indeed, for each reference that was covered, we provided an overview that highlighted the type of evidence as well as its limitations.
To show the reader the development and drawbacks of the treatment of BM, the authors listed the evidence in chronological order, from oldest to newest.
Example:
“In the early 2000s, the Radiation Therapy Oncology Group (RTOG)-9508 trial evaluated the SRS boost after WBRT in 333 patients with 1-3 BMs (19). This trial was designed to detect a 50% increase in median OS in the SRS boost group stratified by BM number (1 vs. 2-3) and RPA (I vs. II).” Line 118.
“A meta-analysis of these trials was published in 2012, evaluating WBRT + SRS vs. WBRT (n = 2) and SRS vs. SRS + WBRT (n = 3)” Line 146.
“Hong et al. reported a randomized phase III trial that included 215 patients with 1-3 melanoma BM (23). The patients underwent local therapy in the form of SRS or surgery, after which they were randomly assigned to either WBRT or observation”. Line 189
“The JLGK0901 prospective observational study, which included 1194 fit patients with 1-10 BM who have been treated with SRS alone, demonstrated no inferiority in OS between those with 2-4 vs. 5-10 BM, with the caveat of a total cumulative intracranial tumor volume ranging from 0.02 to 13.9 cc (≤15 cc).” Line 282
“Hughes et al. published the results of multi-institutional study that included 2089 patients who underwent SRS alone for 1, 2-4, and 5–15 (n=212) BM. Line 306
“Chen et al. conducted a meta-analysis in 2021 that evaluated the efficacy and safety of SRS in a population of 1,446 patients (1,590 BSM) across 32 retrospective studies (excluding the 547 patients from the IGKRF cohort)” Line 466.
- In line 74, 'SBRT' should be spelled out as 'stereotactic body radiotherapy.' Additionally, 'SRS' was spelled out twice, in line 57 and 61, which can be streamlined for clarity. The phrase 'stereotactic radiosurgery' was redundant in line 100.
Response:
The authors thank the reviewers for these remarks. We made the modification.
- The statement 'a formal literature search was not performed' may diminish the scientific rigor and persuasiveness of this review.
Response:
The authors thank the reviewers for their remarks. In response to the reviewers' comments, the authors rewrote the materials and methods section:
Materials and methods:
A narrative literature review was conducted using the databases PubMed, Embase, Google Scholar, and Cochrane. Authors searched the databases until August 2023. Brain metastases, limited brain metastases, multiple brain metastases, brainstem metastases, radiosurgery, stereotactic fractionated radiotherapy, radiotherapy, consensus, expert recommendations, systemic therapy, immune checkpoint inhibitors, and immune radiotherapy were among the keyword combinations that were used. Following it, the results were filtered, and the authors examined therapeutic interventional studies, prospective and retrospective trials, that report on neurocognition, performance status (PS), quality of life (QoL), autonomy in daily activities, toxicity, intracranial progression-free survival (PFS), local control, distant brain control, neurocognitive performance, and PS preservation. Articles that were not relevant to the subject matter of our review were excluded. The authors prioritized prospective trials and meta-analyses to be described in the main text. 993 articles were identified as matching our search terms. After applying filters, 137 documents were identified. Following a prioritization process, 95 were chosen for inclusion.
The authors want to emphasize the broad spectrum of issues of brain metastases that were covered in our review, which could be written as several independent reviews.
For example, SRS and SRT in the management of a limited number of BM (1-4): an accepted treatment; or SRS and SRT in the treatment of multiple BM (> 4): a matter of debate; or brainstem lesions: reaching the limits of SRT.
Reviewer 2 Report
Comments and Suggestions for Authors
Stereotactic radiosurgery (SRS) and Stereotactic fractionated radiotherapy (SFR) in the management of brain intracranial metastases. The authors have made a good effort in presenting a literature review of the existing outcomes for treatment of BMs.
We suggest the following revisions in the article.
1. The authors need to highlight the specific methodology followed. The type of literature search e.g. systematic review or scoping review? The specific inclusion/ exclusion criteria for articles have not been mentioned by the authors.
2. Terminology should be clear and consistent. SRS is defined by organized radiation oncology and neurosurgery as being from 1 to 5 sessions/fractions, and the authors should refer to it as such
3. The authors have emphasized the results of RTOG-9508 trial throughout the initial segment of the paper. They should try to compare smaller groups of literature with similar treatment paradigm and common outcome of interests. A cumulative table of outcomes may be sufficient and if possible, an analysis would be desirable.
The authors may try to combine similar types of evidenc into one section, ege.b. a- meta analysis first and smaller retrospective studies later. Either way, an organized approach to presenting this exhaustive literature review is needed.
4. The discussion has a mixture of results for different metastasis. We suggest that the authors stratify the discussion into sections with organ specific/ disease specific results eg- all melanoma outcomes may be discussed at once within 1-3 lesions group etc.
5. The authors should add a section on the maximum number of mets treated by SRS. It is plausible that several centres would have reported isolated cases of >30 mets treated with SRS at a single session and subsequently at follow up. This multiplicity of treatment sessions provided by SRS, needs to be highlighted when comparing its superiority over WBRT.
6. Conclusion of the article is generalized. There is no specificity on the highlights from the literature search.
Decision: Major revision
Author Response
Reviewer #2:
Stereotactic radiosurgery (SRS) and Stereotactic fractionated radiotherapy (SFR) in the management of brain intracranial metastases. The authors have made a good effort in presenting a literature review of the existing outcomes for treatment of BMs.
We suggest the following revisions in the article.
- The authors need to highlight the specific methodology followed. The type of literature search e.g. systematic review or scoping review? The specific inclusion/ exclusion criteria for articles have not been mentioned by the authors.
Response:
The authors thank the reviewers for their remarks. In response to the reviewers' comments, the authors rewrote the materials and methods section:
Materials and methods:
A narrative literature review was conducted using the databases PubMed, Embase, Google Scholar, and Cochrane. Authors searched the databases until August 2023. Brain metastases, limited brain metastases, multiple brain metastases, brainstem metastases, radiosurgery, stereotactic fractionated radiotherapy, radiotherapy, consensus, expert recommendations, systemic therapy, immune checkpoint inhibitors, and immune radiotherapy were among the keyword combinations that were used. Following it, the results were filtered, and the authors examined therapeutic interventional studies, prospective and retrospective trials, that report on neurocognition, performance status (PS), quality of life (QoL), autonomy in daily activities, toxicity, intracranial progression-free survival (PFS), local control, distant brain control, neurocognitive performance, and PS preservation. Articles that were not relevant to the subject matter of our review were excluded. The authors prioritized prospective trials and meta-analyses to be described in the main text. 993 articles were identified as matching our search terms. After applying filters, 137 documents were identified. Following a prioritization process, 95 were chosen for inclusion.
The authors want to emphasize the broad spectrum of issues of brain metastases that were covered in our review, which could be written as several independent reviews.
For example, SRS and SRT in the management of a limited number of BM (1-4): an accepted treatment; or SRS and SRT in the treatment of multiple BM (> 4): a matter of debate; or brainstem lesions: reaching the limits of SRT.
- Terminology should be clear and consistent. SRS is defined by organized radiation oncology and neurosurgery as being from 1 to 5 sessions/fractions, and the authors should refer to it as such.
Response:
The authors thank the reviewers for their remarks. We made the correction.
- The authors have emphasized the results of RTOG-9508 trial throughout the initial segment of the paper. They should try to compare smaller groups of literature with similar treatment paradigm and common outcome of interests. A cumulative table of outcomes may be sufficient and if possible, an analysis would be desirable.
Response:
The reviewers' comments are appreciated by the authors. The Radiation Therapy Oncology Group (RTOG)-9508 trial evaluated the SRS boost after WBRT in 333 patients with 1-3 BMs. Additionally, the authors reviewed the second retrospective analysis of the RTOG-9508 trial.
Putting together evidence for this patient population through comparisons of smaller groups of literature would not be valuable. In fact, the majority of international guidelines endorse SRS alone as the accepted treatment modality for patients with adequate performance status and 1-4 intact BM (Rhun EL et.al 2012; Soffietti R et al. 2017; Chao ST et al. 2018; Gondi V et al. 2022).
- The authors may try to combine similar types of evidenc into one section, ege.b. a- meta analysis first and smaller retrospective studies later. Either way, an organized approach to presenting this exhaustive literature review is needed.
Response:
The authors thank the reviewers for their remarks. Concerning the evidence class, the authors consistently specified the study type (retrospective, RCT, meta-analysis) and provided crucial information (design, inclusion criteria, population characteristics). Indeed, for each reference that was covered, we provided an overview that highlighted the type of evidence as well as its limitations.
To show the reader the development and drawbacks of the treatment of BM, the authors listed the evidence in chronological order, from oldest to newest.
Example:
“In the early 2000s, the Radiation Therapy Oncology Group (RTOG)-9508 trial evaluated the SRS boost after WBRT in 333 patients with 1-3 BMs (19). This trial was designed to detect a 50% increase in median OS in the SRS boost group stratified by BM number (1 vs. 2-3) and RPA (I vs. II).” Line 118.
“A meta-analysis of these trials was published in 2012, evaluating WBRT + SRS vs. WBRT (n = 2) and SRS vs. SRS + WBRT (n = 3)” Line 146.
“Hong et al. reported a randomized phase III trial that included 215 patients with 1-3 melanoma BM (23). The patients underwent local therapy in the form of SRS or surgery, after which they were randomly assigned to either WBRT or observation”. Line 189
“The JLGK0901 prospective observational study, which included 1194 fit patients with 1-10 BM who have been treated with SRS alone, demonstrated no inferiority in OS between those with 2-4 vs. 5-10 BM, with the caveat of a total cumulative intracranial tumor volume ranging from 0.02 to 13.9 cc (≤15 cc).” Line 282
“Hughes et al. published the results of multi-institutional study that included 2089 patients who underwent SRS alone for 1, 2-4, and 5–15 (n=212) BM. Line 306
“Chen et al. conducted a meta-analysis in 2021 that evaluated the efficacy and safety of SRS in a population of 1,446 patients (1,590 BSM) across 32 retrospective studies (excluding the 547 patients from the IGKRF cohort)” Line 466.
- The discussion has a mixture of results for different metastasis. We suggest that the authors stratify the discussion into sections with organ specific/ disease specific results eg- all melanoma outcomes may be discussed at once within 1-3 lesions group etc.
Response:
The reviewers' comments are appreciated by the authors. Agreement was reached that the discussion comprised a wide range of findings about distinct tumors. As stated in line 213, the majority of the BM trials, examined various varieties of histological cancer, with lung and breast cancer being the most prevalent and melanoma comprising less than 10% of the cohort.
Additionally, regarding the systemic treatment, the majority of BM trials omitted to provide exhaustive details. As indicated in line 217, immune checkpoint inhibitors were used in less than 10% of participants in the Hong et al. melanoma BM trial, and 77% of participants lacked systemic treatment at baseline. Nowadays, targeted therapies and immune checkpoint inhibitors have strengthened extra- and intracranial control in several malignancies: "The combination of SRS-SRT with systemic treatments" is the title of the paragraph that the authors included in this context.
The only RCT that included melanoma BM was referenced and discussed by the authors (line 205). Table 1. is about the other undergoing trial. In conclusion, authors think that classified the discussion into sections with organ specific/ disease specific results is not appropriate. As authors said in the introduction “Significant challenges in BM patients need to be highlighted, such as the wide range of tumor-patient characteristics, the nature of metastasis-directed therapies and the integration of innovative and effective systemic treatment” Line81.
We added:
Introduction part:
[..]
“Patients with BM are a diverse group that has different primary tumors, treatment modalities, signs and symptoms, and life expectancies. Therefore, the optimal management is a complex process that is influenced by several factors, such as their performance status (PS), the type of cancer, the size quantity velocity of BM, the availability of drugs that may effectively penetrate the central nervous system. The process of designing appropriate clinical trials for patients with BM remains a challenging undertaking. In 2023, the collaborative workshop organized by the National Cancer Institute highlighted the importance of establishing an agreement about reproducible and coordinated clinical investigation endpoints in the field of BM research.
[..]
SRS and SRT in the management of a limited number of BM (1-4): an accepted treatment, part:
[..]
Histology-specific investigations are essential in the examination of BM. However, conducting them can be challenging due to the presence of several potential factors that might influence the findings obtained from patients. Further study will be needed to determine the use and timing of SRS-SRT as a means of treatment for patients with multiple BM as the efficacy of molecular and immunotherapy growth. In addition, previously treated brain metastases should also be thoroughly investigated in specific trial.
[..]
In summary, the authors argue the opinion that it is unsuitable to divide the discussion into sections based on disease-specific outcomes.
- The authors should add a section on the maximum number of mets treated by SRS. It is plausible that several centres would have reported isolated cases of >30 mets treated with SRS at a single session and subsequently at follow up. This multiplicity of treatment sessions provided by SRS, needs to be highlighted when comparing its superiority over WBRT.
Response:
There is insufficient evidence to support the claim that the SRS is preferable to the WBRT for >30 BM. Trials are underway. Table 1 presents a comprehensive overview of the principal trial that evaluated the efficacy of SRS-SRT in conjunction with systemic therapies. The authors made a paragraph titled "SRS and SRT in the treatment of multiple BM (> 4): a matter of debate." for this purpose and wrote “ The most effective treatment for multiple brain metastases has become the subject of increased debate in the past decade and is an investigation of significant epidemiologic relevance. Several advanced technologies, such as patient setup, target localization, treatment planning, and delivery, have changed and personalized the way radiation is delivered to BM patients. Although there are not enough data from trials to show that WBRT is a better option than SRS alone for patients with multiple BM, it has long been accepted practice that these patients should receive WBRT.
[..]
RCTs are currently underway in patients with 5-15 BM (Table.1). The most effective way for physicians to treat patients with multiple BM will be debated until these trials are published. Nevertheless, in recent years, SRS has become increasingly common in patients with multiple BM
[..]
[..]
This salvage treatment emphasizes the importance of close MRI follow-up when treating with SRS alone, as 50% of patients will develop new BM and distant brain failure increases with time (46)
[..]
Nowadays, WBRT is no longer the traditional approach for BM patients as the treatment paradigm is rapidly changing and advancing (12). The rapid pace of technical advancements in RT has outpaced the availability of clinical evidence. Consequently, there is an urgent need for high-quality level 1 evidence to provide a more accurate understanding of the role of SRS in patients with multiple BM.
[..]
Nevertheless, if SRS-SRT is chosen as a treatment option for multiple BM, it is essential to emphasize the importance of close follow-up with sequential MRI (at least once every two to three months during the first year after treatment or whenever there is a possible neurological progression). In fact, SRS-SRT does not have a prophylactic purpose compared to WBRT, and close surveillance is necessary for the early diagnosis of new BM. The SRS-SRT technique effectively delivers a highly targeted and heterogeneous dose specifically to the metastatic site, while ensuring tiny exposure of healthy brain and a low mean brain dose. In contrast to WBRT, this treatment can be repeated as needed and is more convenient for patients, as SRS is usually delivered in only 1 to 5 sessions (compared to 10 fractions in the case of WBRT).
[..]
In the introduction part authors also said “It should be emphasized that SRS-SRT requires resources, training, and widely available equipment (e.g. magnetic resonance imaging) with accurate and reliable systems”. Line 79.
Reviewer 3 Report
Comments and Suggestions for Authors
This review paper delves into Stereotactic Radiosurgery (SRS) and Stereotactic Fractionated Radiotherapy (SFR) for the treatment of brain metastases, raising several concerns throughout the analysis:
- Abstract: The primary concern lies in the lack of clarity regarding the aim of this review. The authors need to articulate the purpose behind presenting an overview of the current landscape of SRS and SFR for treating brain metastases. Readers should be informed about the key takeaways they can expect from this review.
- Introduction: While the authors touch upon the potential benefits and limitations of SRS and SFR in the last paragraph of the introduction, there is a need for a more comprehensive discussion. The authors should not only describe the current state but also elaborate on the future direction, prospects, and a potential roadmap for these treatment modalities.
- Introduction: The authors exclusively mention Gamma knife as a treatment unit for SRS and SFR, overlooking other modalities like Linac-based and CyberKnife. Contemporary options, such as Linac-based treatments (e.g., Asnasshari et al Phys Med 2013;29:350) and CyberKnife (e.g., Sio et al JACMP 2014;15:14), should be incorporated into the discussion.
- Materials and Methods L93-102: Regarding the literature survey, it is recommended that the authors provide details on the number of papers initially searched and how many were subsequently filtered out during the data mining process.
- Table 1: The presentation of Table 1 needs improvement in terms of format and readability. Additionally, it would be beneficial to assign reference numbers to the first column, facilitating cross-referencing with the Reference section.
- Future Direction and Perspective: A critical omission in the paper is the absence of a section analyzing the findings from the literature survey to identify the future direction of SRS and SFR for brain metastases. The authors should include a dedicated section addressing the potential advancements and trends emerging from their analysis.
Comments on the Quality of English Language
No problem in English
Author Response
Review 3#:
This review paper delves into Stereotactic Radiosurgery (SRS) and Stereotactic Fractionated Radiotherapy (SFR) for the treatment of brain metastases, raising several concerns throughout the analysis:
- Abstract: The primary concern lies in the lack of clarity regarding the aim of this review. The authors need to articulate the purpose behind presenting an overview of the current landscape of SRS and SFR for treating brain metastases. Readers should be informed about the key takeaways they can expect from this review.
Response:
The authors thank the reviewers for their remarks. Authors change the last paragraph:
[..]
This review aims to comprehensively explore SRS and SRT as treatments for BM, covering clinical considerations in their application (e.g., patient selection and eligibility), managing limited and multiple intact BM, ad-dressing brain stem metastases, exploring combination therapies with systemic treatments, and considering the health economic perspective.
- Introduction: While the authors touch upon the potential benefits and limitations of SRS and SFR in the last paragraph of the introduction, there is a need for a more comprehensive discussion. The authors should not only describe the current state but also elaborate on the future direction, prospects, and a potential roadmap for these treatment modalities.
Response:
The authors thank the reviewers for their comments. We added a paragraph as requested.
[..]
To allow to conduct comprehensive comparisons, stratified BM patients must be eligible to participate in clinical trials. Patients with BM are a diverse group that has different primary tumors, treatment modalities, signs and symptoms, and life expectancies. Therefore, the optimal management is a complex process that is influenced by several factors, such as their performance status (PS), the type of cancer, the size quantity velocity of BM, the availability of drugs that may effectively penetrate the central nervous system. The process of designing appropriate clinical trials for patients with BM remains a challenging undertaking.
[..]
- Introduction: The authors exclusively mention Gamma knife as a treatment unit for SRS and SFR, overlooking other modalities like Linac-based and CyberKnife. Contemporary options, such as Linac-based treatments (e.g., Asnasshari et al Phys Med 2013;29:350) and CyberKnife (e.g., Sio et al JACMP 2014;15:14), should be incorporated into the discussion.
Response:
The authors acknowledge the reviewers' comment and appreciate the feedback. Upon careful consideration, the authors agree that two of the proposed references, namely Sio et al. and Asnasshari et al., may not align well with the subject matter of the review. Sio et al. focuses on the influence of patient's physiologic factors and immobilization choice with stereotactic body radiotherapy for upper lung tumors, while Asnasshari et al. examines dosimetric parameters of two multi-leaf collimator (MLC) systems, specifically Elekta's "Synergy S" linear accelerator and Radionics micro-MLC (MMLC).
- Materials and Methods L93-102: Regarding the literature survey, it is recommended that the authors provide details on the number of papers initially searched and how many were subsequently filtered out during the data mining process.
Response:
The authors thank the reviewers for their remarks. In response to the reviewers' comments, the authors rewrote the materials and methods section:
Materials and methods:
A narrative literature review was conducted using the databases PubMed, Embase, Google Scholar, and Cochrane. Authors searched the databases until August 2023. Brain metastases, limited brain metastases, multiple brain metastases, brainstem metastases, radiosurgery, stereotactic fractionated radiotherapy, radiotherapy, consensus, expert recommendations, systemic therapy, immune checkpoint inhibitors, and immune radiotherapy were among the keyword combinations that were used. Following it, the results were filtered, and the authors examined therapeutic interventional studies, prospective and retrospective trials, that report on neurocognition, performance status (PS), quality of life (QoL), autonomy in daily activities, toxicity, intracranial progression-free survival (PFS), local control, distant brain control, neurocognitive performance, and PS preservation. Articles that were not relevant to the subject matter of our review were excluded. The authors prioritized prospective trials and meta-analyses to be described in the main text. 993 articles were identified as matching our search terms. After applying filters, 137 documents were identified. Following a prioritization process, 95 were chosen for inclusion.
The authors want to emphasize the broad spectrum of issues of brain metastases that were covered in our review, which could be written as several independent reviews.
For example, SRS and SRT in the management of a limited number of BM (1-4): an accepted treatment; or SRS and SRT in the treatment of multiple BM (> 4): a matter of debate; or brainstem lesions: reaching the limits of SRT.
- Table 1: The presentation of Table 1 needs improvement in terms of format and readability. Additionally, it would be beneficial to assign reference numbers to the first column, facilitating cross-referencing with the Reference section.
Response:
The authors appreciate the reviewers' feedback regarding Table 1. We acknowledge the need for improvement in terms of format and readability. In the revised manuscript, we will assign reference numbers to the first column to facilitate cross-referencing with the Reference section, providing a more organized and accessible presentation of the information. The final layout will be crafted in accordance with the journal's policy.
- Future Direction and Perspective: A critical omission in the paper is the absence of a section analyzing the findings from the literature survey to identify the future direction of SRS and SFR for brain metastases. The authors should include a dedicated section addressing the potential advancements and trends emerging from their analysis.
Response:
The authors appreciate the reviewers' insights, acknowledging. In response, the authors provided an in-depth discussion of potential advancements and emerging trends, specifically in the "SRS and SRT in the treatment of multiple BM (> 4): a matter of debate" section. Additionally, Table 1 was incorporated. It is important to note that the ongoing trials will play a key role in shaping the future direction.
As the authors said in the simple summary “This review highlights the evidence and the emerging role of SRS-SRT in patients diagnosed with intact intracranial metastases.” Also at the end of the introduction: “The aim of this review is to describe the potential benefits and limitations of stereotactic radiosurgery and stereotactic radiotherapy in the treatment of brain metastases”.
Authors added:
Health economic perspective:
[..]
So far, evidence for SRS alone in patients with BM has come from studies in which patients were mostly included and accepted based on the BM number. Even though the amount of BM has been shown to be important, it is not the only one. Factors such the location (e.g., brain stem, see below), the CITV, the kinetic, and the systemic treatment should also influence outcomes but have not been significantly investigated.
The variability observed in this context has resulted in a lack of consistency when comparing the effects of treatment across various studies. As an example, the Response Assessment in Neuro-Oncology Brain Metastases (RANO-BM) criteria should be used instead of the RECIST criteria to accurately evaluate both local and distant recurrence. Importantly, mapping the recurrence's location with the radiotherapy isodose must be mandatory and provided. A range and combination of cognitive function monitoring assessments, functional status, toxicity, health-related QoL, and patient-reported outcomes (PROMs) that are specific to the side effects of radiation should all be included in the study's results. The Brain Symptom Impact Questionnaire (BASIQ) and the Functional Assessment of Cancer Therapy general and brain (FACT-G; FACT-B) were the sole two PROMs which were explicitly established in the BM patient (78). A complete understanding of the long-term evolution of patients with BM is necessary. An area requiring effort is the development of instruments developed specifically for BM patients that may be easily implemented by multiple organizations into everyday clinical practice.
[..]
Round 2
Reviewer 1 Report
Comments and Suggestions for Authors
I enjoyed reading the revised manuscript. A few points need to be clarified.
Line 216: How should we understand the sentence, “missing the SRS to the surgical cavity resulting in significantly higher local failure, which could potentially contribute to the development of distant brain metastases”?
Line 473: There seems to be a missing "C" in "NSCL."
Line 582: It should be “>” instead.
Line 658: There seems to be a missing "as" in "such as."
Reviewer 3 Report
Comments and Suggestions for Authors
The authors addressed most of my concerns.
Comments on the Quality of English Language
NA